# Continuous co-prescription of rebamipide prevents upper gastrointestinal bleeding in NSAID use for orthopaedic conditions: A nested case-control study using the LIFE Study database

**Satoshi Yamate**[1], **Chieko Ishiguro**[2], **Haruhisa Fukuda**[3,4], **Satoshi Hamai**[1]*, **Yasuharu Nakashima**[1]

1 Department of Orthopaedic Surgery, Graduate School of Medical Sciences, Kyushu University, Fukuoka, Japan, 2 Department of Data Science, Section of Clinical Epidemiology, Center for Clinical Sciences, National Center for Global Health and Medicine, Tokyo, Japan, 3 Department of Health Care Administration and Management, Graduate School of Medical Sciences, Kyushu University, Fukuoka, Japan, 4 Center for Cohort Studies, Graduate School of Medical Sciences, Kyushu University, Fukuoka, Japan

* shamai0220@gmail.com

**Data Availability Statement:** The data used in this study were acquired under agreements between Kyushu University and the participating

## Abstract

### Background

Rebamipide has been widely co-prescribed with non-steroidal anti-inflammatory drugs (NSAIDs) in Japan for decades. This study aimed to evaluate the effectiveness of rebamipide in preventing upper gastrointestinal bleeding in new users of NSAIDs without risk factors of NSAID-induced ulcers other than age.

### Methods

A nested case-control study was conducted using medical claims data of 1.66 million inhabitants of 17 municipalities participating in Japan's Longevity Improvement & Fair Evidence study. The cohort entry ($t_0$) corresponded to a new user of NSAIDs for osteoarthritis or low back pain. Patients with risk factors of NSAID-induced ulcers other than age were excluded. Cases were defined as patients who underwent gastroscopy for upper gastrointestinal bleeding (occurrence date was defined as index date). A maximum of 10 controls were selected from non-cases at the index date of each case by matching sex, age, follow-up time, and type and dosage of NSAIDs. Exposure to rebamipide was defined as prescription status from $t_0$ to index date: Non-user (rebamipide was not co-prescribed during the follow-up period), Continuous-user (rebamipide was co-prescribed from $t_0$ with the same number of tablets as NSAIDs), and Irregular-user (neither Non-user nor Continuous-user). Conditional logistic regression analysis was conducted to estimate each category's odds ratio compared to non-users.

municipalities, which stipulate that the data can only be used by authorized research institutions and cannot be shared with third parties. However, research institutions that have entered into agreements with the authorized research group in Kyushu University may access the data. Please contact the Joint Research Department of Kyushu University (ijkkyoudou@jimu.kyushu-u.ac.jp) regarding data access.

**Funding:** This work was supported by the Japan Society for the Promotion of Science KAKENHI (Grant Numbers JP19K21590 and JP20H00563) and by Japan Science and Technology Agency FOREST (Fusion Oriented REsearch for disruptive Science and Technology) program (Grant Number JPMJFR205J). The funders had no role in study design, data collection and analysis, decision to publish, or preparation of the manuscript.

**Competing interests:** Authors with competing interests I have read the journal's policy and the authors of this manuscript have the following competing interests:Yasuharu Nakashima received research grants from Chugai Pharmaceutical Co., Ltd., AYUMI Pharmaceutical Corporation, EA Pharma Co., Ltd., KYOCERA Corporation, and Zimmer Biomet G.K. All other authors have no conflicts of interest to disclose. This does not alter our adherence to PLOS ONE policies on sharing data and materials.

## Findings

Of 67,561 individuals who met the inclusion criteria, 215 cases and 1,516 controls were selected. Compared with that of Non-users, the odds ratios and 95% confidence interval were 0.65 (0.44–0.96) for Continuous-users and 2.57 (1.73–3.81) for Irregular-users.

## Conclusions

Continuous co-prescription of rebamipide significantly reduced the risk of upper gastrointestinal bleeding in an Asian cohort of new users of NSAIDs with osteoarthritis or low back pain without risk factors other than age.

## Introduction

Rebamipide is a peptic ulcer protective drug developed in Japan [1] and is the most popular drug used in Japan for this purpose. According to the government's annual all-counts survey, 2.5 billion tablets (0.25 million kg) of rebamipide, marketed by 24 pharmaceutical companies, were prescribed from April 2021 to March 2022 [2]. Rebamipide promotes the synthesis of prostaglandins [3] and secretion of gastric mucus [4], thereby inhibiting the production of reactive oxygen radicals and inflammatory cytokines and suppressing the activity of leukocytes [5]. In East Asia, rebamipide is widely prescribed as a prophylaxis for non-steroidal anti-inflammatory drug (NSAID)-induced ulcers [6]. However, current guidelines in Japan [7] and abroad [8] do not describe rebamipide's effectiveness, usage, or efficacy.

The Japanese guideline [7] recommends the co-prescription of proton-pump inhibitors (PPIs) in preventing NSAID-induced ulcers; however, Japanese insurance does not cover using PPIs as prophylaxis for NSAID-induced ulcers. The effectiveness of PPIs in preventing NSAID-induced ulcers has been reported in numerous randomized controlled trials [9–13] and is also recommended by worldwide guidelines for moderate or high-risk patients who have any of the following risk factors: age over 65 years, previous history of an uncomplicated ulcer, or concurrent use of aspirin, antiplatelet drugs, corticosteroids, or anticoagulant agents [8]. However, their effectiveness in low-risk patients without risk factors remains unclear. Recently, there have been several reports linking PPIs with adverse drug reactions, such as dementia [14], bone fractures [15], prosthetic joint infection [16], and severe clinical outcomes of coronavirus disease (COVID-19) [17]. However, it is also essential to consider potential bias in reporting these associations [18–20]. On the other hand, rebamipide prevents ulcers without affecting gastric acid secretion [1], is known to have very few adverse drug reactions [5, 21], and could be considered as an alternative to PPIs as a prophylactic agent for patients at low risk of NSAID-induced ulcers [6].

The prevention of NSAID-induced ulcers can be established from the initial prescription of NSAIDs, such as in an orthopaedic outpatient clinic; however, several of these prescriptions fail to adhere to the guidelines for preventing ulcers [22]. In our study, we focused on new users of NSAIDs with low back pain or joint pain caused by osteoarthritis, which are typical orthopaedic diseases [23–26], to evaluate the effectiveness of co-prescription of rebamipide in preventing NSAID-induced ulcers. Our investigation aimed to answer the question, "Does rebamipide contribute to preventing upper gastrointestinal bleeding in new users of NSAIDs for osteoarthritis or low back pain and who are at low risk of NSAID-induced ulcers?" We hypothesized that continuous co-prescription of rebamipide would have a preventive effect on NSAID-induced ulcers.

## Materials and methods

### Study design

Using routinely collected medical claims data, this observational study using a nested case-control study design [27–29] was conducted following the STROBE (Strengthening the Reporting of Observational Studies in Epidemiology) guidelines [30], RECORD (Reporting of Studies Conducted Using Observational Routinely-Collected Health Data) Statement [31], and ethical standards of the Declaration of Helsinki.

We used a nested case-control study design to account for many covariates of bleeding gastric ulcers, such as the type and dosage of NSAIDs prescribed, treatment duration, comorbidities, and co-prescribed drugs that may cause ulcers. In addition, changes in the type or dosage of NSAIDs prescribed during the study period were considered an essential confounding factor associated with the occurrence of events. Therefore, the study design, which could incorporate the information on the dosage and type of NSAIDs prescribed, was considered appropriate to assess the effect of rebamipide.

### Setting

We used medical claims data for the analysis as we considered it useful for tracking patients across specialties in the real world. We reviewed the medical claims data collected from April 1, 2013, to December 31, 2020, of 1.66 million inhabitants of 17 municipalities participating in the Longevity Improvement & Fair Evidence (LIFE) Study [32–34], a research database project by Kyushu University, Fukuoka, Japan, and approved by the Kyushu University Institutional Review Board for Clinical Research (Registration number: 22114–04).

The LIFE Study collected medical claims data from two public insurance systems: Japan's National Health Insurance System for individuals aged 0–74 years and Latter-Stage Older Persons Health Care System for individuals aged ≥75 years and those aged 65–74 years with specific diseases. The former enrolls 26.3% of all citizens aged 0 to 74, of which 68.5% of all citizens aged 65 to 74 are enrolled, and the latter is in principle enrolled by all citizens aged 75 and older, thus these two insurance policies cover the majority of the elderly in Japan [32, 35]. The claims data included recorded diagnoses based on the International Classification of Disease, Tenth Revision (ICD-10) codes, drug prescriptions, and medical treatment during clinical encounters, collected by five researchers using pre-delineated forms of LIFE Common Data Model [32].

The review board waived the requirement for informed consent due to the study's retrospective nature and because all records were de-identified and fully anonymized before our access for analysis. The data were accessed for research purposes from May 11, 2022, to March 24, 2023. The authors did not have access to information that could identify individual participants during or after data collection.

### Participants

**Definition of cohort.** The cohort included patients who were prescribed NSAIDs, including selective cyclooxygenase-2 (COX-2) inhibitors (S1 Table), in the calendar month when the ICD-10 codes for osteoarthritis (M16-19) and back pain (M54) were recorded. The calendar date of the first NSAID prescription was defined as $t_0$. Considering that insurance payers often do not allow prescriptions to exceed 90 days, therefore, the time window (T.W.) from 90 days before $t_0$ to 1 day before $t_0$ was used for defining exclusion criteria.

The exclusion criteria were as follows: (1) patients without medical records before the T.W., (2) patients who received NSAIDs without an ICD-10 diagnosis of osteoarthritis or back pain

during the T.W., (3) patients with a history of upper gastrointestinal ulcers (ICD-10 codes: K25-28 and K922) or diagnosis related to *Helicobacter pylori* infection (ICD-10 codes: K294, K296, A048, A498, and B980), (4) patients who received drugs associated with the risk of gastric ulcer development or associated with bleeding [7] (such as anticoagulants, antiplatelet agents, steroids, or bisphosphonates) prescribed during the T.W., (5) patients with a history of upper gastrointestinal endoscopy before $t_0$, (6) patients with ulcer treatment or prophylaxis (rebamipide, PPI, misoprostol, H2 receptor antagonists, or other gastroprotective medicines) during the T.W., and (7) patients with ulcer treatment or prophylaxis other than rebamipide at $t_0$. These inclusion and exclusion criteria aimed to select patients who were prescribed NSAIDs alone or NSAIDs with rebamipide for the first time to treat joint or back pain, with no risk factors [8] other than age. We excluded cases in which rebamipide was used to treat *H. pylori*-associated gastritis [36–38], as our objective was to assess the effectiveness of rebamipide for the prevention and not for the treatment of gastric ulcers.

The follow-up period encompassed the prescription date with a 180-day gap and a 180-day grace period for NSAIDs. Therefore, if any NSAIDs were prescribed within 180 days of the last prescription date, it was considered as continued treatment. Patients who received high-risk drugs (anticoagulants, antiplatelet agents, steroids, or bisphosphonates) after $t_0$ were censored on that previous day. Therefore, only data from the period until the prescription of high-risk drugs were used for the analysis.

**Definition of cases and controls.** In a nested case-control study, cases within the cohort experience the event of interest selected from the follow-up period [29]. In this study, the event of interest was upper gastrointestinal bleeding. Patients were considered as cases if the following two criteria (A) and (B) were met: (A) there was a claim for gastroscopy (treatment code: 160093810) or a hemostatic procedure (treatment code: 150164850) from the day after the first NSAID prescription until 180 days after the last NSAID prescription date within the follow-up period; (B) there was a diagnosis of 'upper gastrointestinal bleeding' (ICD-10 codes: K250, K252, K254, K256, K260, K262, K264, K266, K290, and K922) was required in the same calendar month.

In a nested case-control study, controls are selected from the cohort members who are at risk of the event at the time of each case's occurrence, and these controls can be chosen repeatedly or may later become cases themselves [29]. For each case, controls (maximum 10) were selected from non-cases at the index date of each case by matching (A) sex at $t_0$, (B) age (±5 years) at $t_0$, (C) follow-up (±180 days) from $t_0$ to the index date, (D) the total number of NSAID tablets prescribed, and the total dose of the five types of NSAIDs (loxoprofen, celecoxib, diclofenac, meloxicam, and ibuprofen).

**Grouping.** Patients were grouped based on the prescription status of rebamipide from $t_0$ to the index date: 1) Non-user, 2) Continuous-user, and 3) Irregular-user (Fig 1). Patients without rebamipide prescriptions from $t_0$ to the index date were grouped as Non-users. If patients were continuously prescribed rebamipide from $t_0$ to the index date with the same number of tablets as NSAIDs, they were grouped as Continuous-users. Patients not falling under either Continuous-user or Non-user were grouped as Irregular-users. Therefore, if a patient was prescribed NSAIDs less than three times a day but rebamipide 100 mg three times a day [4], they were classified as an irregular user because the number of tablets did not match.

## Statistical analysis

Conditional logistic regression analysis considering matching factors was conducted to estimate the odds ratio (OR) for the occurrence of NSAID-induced bleeding ulcers to evaluate the association between the use of rebamipide as a prophylaxis for NSAID-induced ulcers and the

**1) Non-user :** Rebamipide was not co-prescribed during the follow-up period.

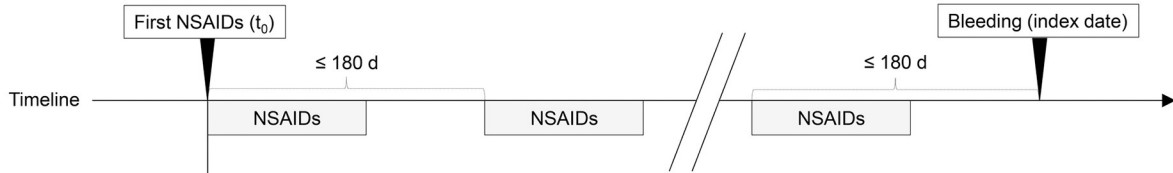

**2) Continuous-user :** Rebamipide was co-prescribed from $t_0$ with the same number of tablets as NSAIDs.

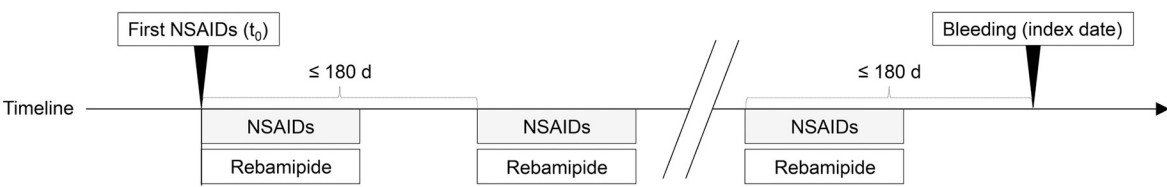

**3) Irregular-user :** Neither Non-user nor Continuous-user.

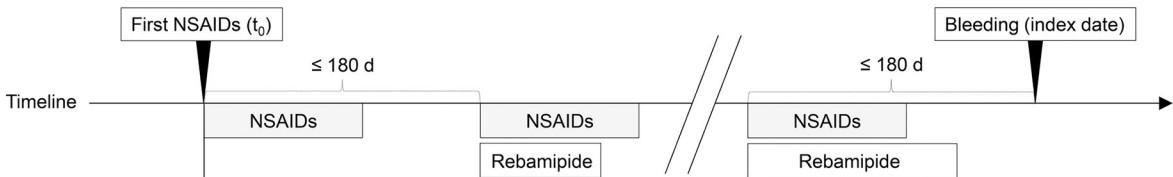

**Fig 1. Grouping based on the prescription status of rebamipide.** Non-user = Rebamipide was not co-prescribed during the follow-up period. Continuous-user = Rebamipide was co-prescribed from $t_0$ with the same number of tablets as NSAIDs. Irregular-user = Neither Non-user nor Continuous-user. NSAIDs, non-steroidal anti-inflammatory drugs.

incidence of upper gastrointestinal bleeding. The reference was defined as no rebamipide prescription during the entire observation period from $t_0$. We performed a subgroup analysis stratifying patients by age, either younger than 65 years of age (*Subgroup analysis 1*) or 65 years of age and older (*Subgroup analysis 2*), of which the latter is a risk factor for NSAID-induced ulcers [8]. In addition, we performed a subgroup analysis based on whether NSAIDs prescribed at $t_0$ were loxoprofen (*Subgroup analysis 3*) or celecoxib (*Subgroup analysis 4*) or diclofenac (*Subgroup analysis 5*). All statistical analyses were performed using R version 4.2.0 (The R Foundation, Vienna, Austria) and RStudio version 2022.02.2 (RStudio, Boston, MA, USA).

## Sensitivity analyses

Considering the possibility that "low-risk" was not well defined in the main analysis [39], we performed three patterns of sensitivity analysis with additional exclusion criteria of patients with an ICD-10-based Charlson Comorbidity Index [40] of ≥1 point (*Sensitivity analysis 1*), ≥2 points (*Sensitivity analysis 2*), or ≥3 points (*Sensitivity analysis 3*).

## Patient and public involvement

In this study, there was no direct involvement from patients or the general public in the research design, conduct, or reporting. However, we shared our study protocol with

participating municipalities. In line with the aim of the LIFE Study database, the analytical results are planned to be fed back to the public through these participating municipalities [32].

## Results

Of the 367,714 patients who were prescribed NSAIDs (including selective COX-2 inhibitors) and had a diagnosis of osteoarthritis or back pain in the same calendar month, 295,950 (80.5%) patients were aged 65 years or older. The most prescribed NSAIDs were loxoprofen (233,575 patients; 63.5%), followed by celecoxib (91,863 patients; 25.0%) and diclofenac (32,817 patients; 8.9%). The co-prescription rate of rebamipide was 36.1% for loxoprofen, 41.2% for celecoxib, and 26.2% for diclofenac, respectively (S2 Table).

After applying the exclusion criteria, 67,561 patients were included in the study cohort (Fig 2). From the cohort, 345 (0.5%) case candidates were identified, and their median age at cohort entry was 77 years (S3 Table). Of the 345 case candidates, 61 had claim codes for hemostatic procedures.

After matching, 215 cases and 1,516 controls were identified; there were no notable differences in patient background between cases and controls (Table 1). Total dose of NSAIDs for the cases used for the control matching factor showed a wide distribution in the quartile range for loxoprofen, celecoxib, and diclofenac. The median time to upper gastrointestinal bleeding after the first prescription of NSAIDs was 134 days (S1 Fig). Regarding the prescription status of rebamipide, 649 patients were classified as continuous-users, 265 patients as irregular-users, and 817 patients as non-users. The OR for the occurrence of upper gastrointestinal bleeding for the continuous-user group was 0.65 (95% confidence interval [CI] 0.44–0.96) compared with that of non-users and 2.57 (95% CI 1.73–3.81) for irregular-users compared with that of non-users (**Table 2**).

For the continuous-user group, the ORs for the occurrence of upper gastrointestinal bleeding were 0.35 (95% CI 0.09–1.32) for the subgroup with patients younger than 65 years of age (S4 Table) and 0.69 (95% CI 0.46–1.04) for the subgroup with patients older than 65 years of age (S5 Table). The results of the subgroup analysis based on NSAIDs prescribed at $t_0$ showed ORs of 0.67 (95% CI 0.42–1.08) for loxoprofen (S6 Table) and 0.58 (95% CI 0.28–1.20) for celecoxib (S7 Table). Sensitivity analyses using the Charlson Comorbidity Index to determine the low-risk factors of NSAID-induced ulcer, other than age, also showed consistent results (S8–S10 Tables).

## Discussion

We found that upper gastrointestinal bleeding significantly decreased when rebamipide was co-prescribed with the same number of tablets from the first dose of NSAID and later continuously prescribed. This result was based on real-world data in orthopaedic patients without risk factors for NSAID-induced ulcers other than age. Using rebamipide for preventing NSAID-induced ulcers, a simple co-prescription pattern for rebamipide may improve adherence in clinical practice. The co-prescription rate of rebamipide with NSAIDs was high in our LIFE database, consistent with that in previous reports [1, 5, 36, 37, 41]. Thus, the study results demonstrate rebamipide's clinical relevance and effectiveness in real-world practice in an Asian cohort.

Our findings also revealed that for irregular-users for whom the total number of rebamipide tablets did not match that of NSAIDs, the occurrence of upper gastrointestinal bleeding was higher than that in non-users. In some cases, rebamipide may be prescribed after the initiation of NSAIDs owing to patients' symptoms, such as stomach pain. Our findings suggest that by re-evaluating the pattern of rebamipide prescription by gastroenterology nonspecialists, early detection of gastric ulcers may be possible.

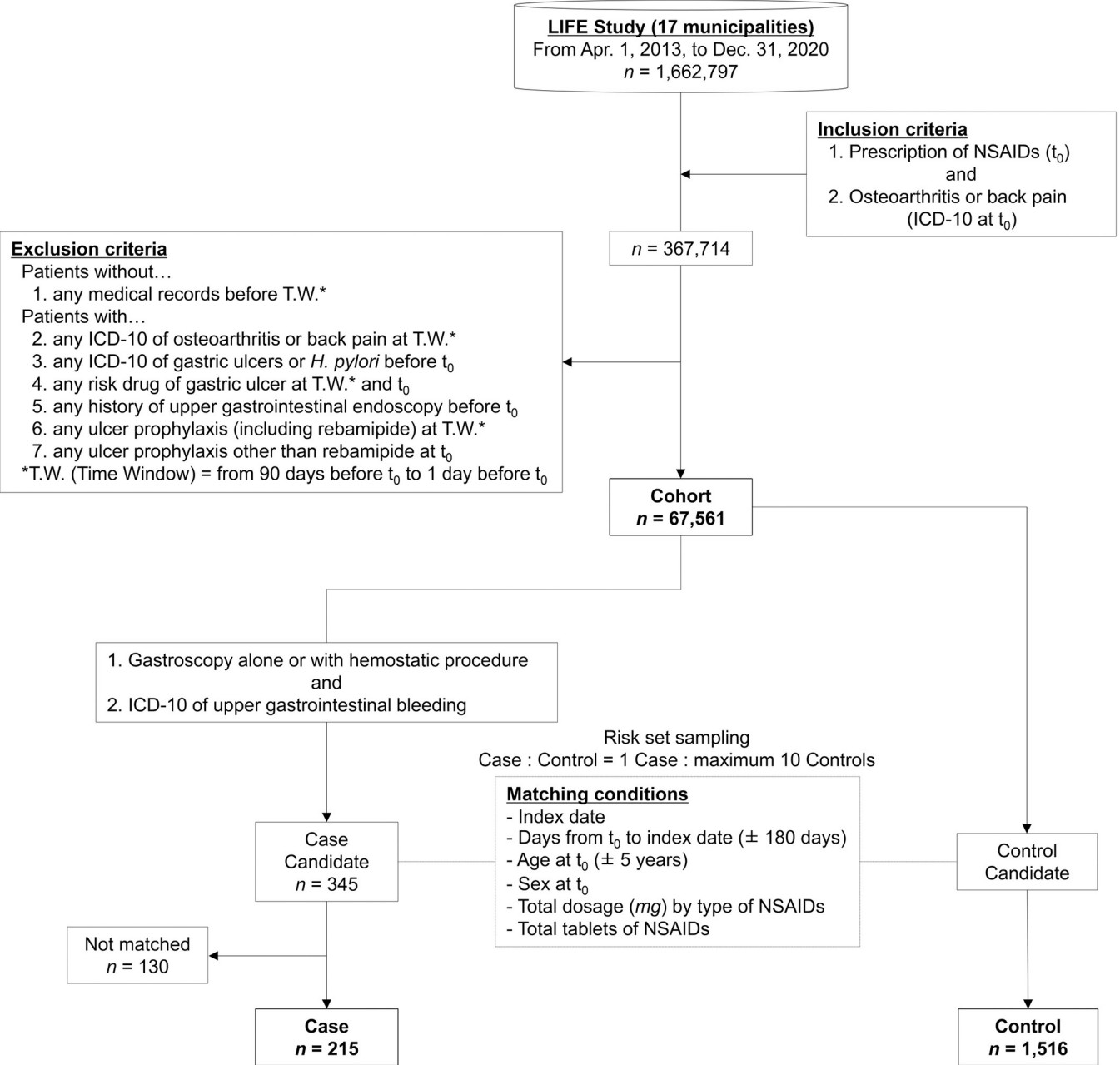

**Fig 2. STROBE diagram illustrating the workflow for patient selection.** STROBE, Strengthening the Reporting of Observational Studies in Epidemiology; LIFE, Longevity Improvement & Fair Evidence; NSAIDs, non-steroidal anti-inflammatory drugs; ICD-10, International Classification of Disease Tenth Revision.

Rebamipide was comparable to misoprostol in reducing the incidence of gastric and duodenal ulcers according to a multicenter, double-blind, randomized parallel-group study by Kim et al. [42] They also found that rebamipide was associated with fewer adverse drug reactions, such as nausea and diarrhea. However, our study found that the actual number of prescriptions is approximately 1/100th of that for rebamipide; this could be because rebamipide has fewer adverse drug reactions than misoprostol [42, 43]. Recently, Lee et al. [6] reported no significant difference in the risk of serious gastrointestinal adverse drug reactions between PPI

**Table 1. Patient characteristics of cases and controls in this study.**

| Variable | Case (*n* = 215) | Control (*n* = 1,516) |
|---|---|---|
| Index date (range)[a] | 11 Sep 2013 to | 11 Sep 2013 to |
| | 11 Nov 2020 | 11 Nov 2020 |
| Days from $t_0$ to index[a] | median 134 | median 159 |
| | IQR 48–377 | IQR 76–292 |
| | range 1–1836 | range 1–1656 |
| Age at $t_0$[a] | median 76.2 | median 75.8 |
| | IQR 69–83 | IQR 69–81 |
| | range 36.7–98.5 | range 32.6–98.8 |
| Age at $t_0$ (no. [%]) | | |
| ≤64 years | 32 (14.9) | 184 (12.1) |
| 65–74 years | 61 (28.4) | 513 (33.8) |
| 75–84 years | 84 (39.1) | 604 (39.8) |
| ≥85 years | 38 (17.7) | 215 (14.2) |
| Sex at $t_0$[a] (no. [%]) | | |
| Male | 107 (49.8) | 764 (50.4) |
| Female | 108 (50.2) | 752 (49.6) |
| Diagnosis at $t_0$ (no. [%]) | | |
| Osteoarthritis of hip | 4 (1.9) | 50 (3.3) |
| Osteoarthritis of knee | 65 (30.2) | 480 (31.7) |
| Osteoarthritis (Others) | 17 (7.9) | 118 (7.8) |
| Back pain | 129 (60.0) | 868 (57.3) |
| Type of NSAIDs at $t_0$ (no. [%]) | | |
| Loxoprofen | 152 (70.7) | 1143 (75.4) |
| Celecoxib | 52 (24.2) | 331 (21.8) |
| Diclofenac | 8 (3.7) | 36 (2.4) |
| Meloxicam | 2 (0.9) | 5 (0.3) |
| Ibuprofen | 1 (0.5) | 1 (0.1) |
| Co-prescription of rebamipide at $t_0$ (no. [%]) | 95 (44.2) | 713 (47.0) |
| Total dose of NSAIDs[a,b] | | |
| Loxoprofen (mg) | (n = 155[c]) | (n = 1,146[c]) |
| | median 600 | median 420 |
| | IQR 330–1,260 | IQR 300–840 |
| | range 60–18,240 | range 60–18,240 |
| Celecoxib (mg) | (n = 52[c]) | (n = 334[c]) |
| | median 1,400 | median 1,400 |
| | IQR 7,00–2,800 | IQR 700–2,800 |
| | range 100–21,000 | range 100–21,000 |
| Diclofenac (mg) | (n = 10[c]) | (n = 38[c]) |
| | median 62.5 | median 50 |
| | IQR 31.25–228.12 | IQR 25–50 |
| | range 25–525 | range 25–525 |
| Meloxicam (mg) | (n = 2[c]) | (n = 5[c]) |
| | median 70 | median 70 |
| | IQR 70–70 | IQR 70–70 |
| | range 70–70 | range 70–70 |
| Ibuprofen (mg) | (n = 1[c]) | (n = 1[c]) |
| | 1,500 | 1,500 |

(*Continued*)

**Table 1.** (Continued)

| Variable | Case (*n* = 215) | Control (*n* = 1,516) |
|---|---|---|
| Total tablets of NSAIDs[a] (median [IQR, range]) | median 10 | median 7 |
| | IQR 6–23 | IQR 5–14 |
| | range 1–304 | range 1–304 |

NSAIDs, non-steroidal anti-inflammatory drugs; IQR, interquartile range (25th percentile–75th percentile).

[a]Variables used for matching.

[b]Data are shown excluding patients with 0 mg, respectively.

[c]Number of patients included in the relevant section after excluding patients with 0 mg.

and rebamipide for preventing NSAID-induced ulcers (adjusted hazard ratio 0.69, 95% CI 0.27–1.76) using the national claims database. Even though the frequency is lower compared to other acid-related disorder treatments, prescribers should be aware of the risk of pulmonary adverse drug reactions associated with rebamipide [44], and leukopenia might occur in rare cases [5], meaning that rebamipide is not entirely risk-free.

Imai et al. [45] found that the reporting ORs for lower gastrointestinal tract injury with loxoprofen and diclofenac in combination with rebamipide were 0.50 (95% CI 0.35–0.71) using the U.S. Food and Drug Administration (FDA) database and 0.43 (95% CI 0.27–0.67) using the Pharmaceuticals and Medical Devices Agency database. Our study on upper gastrointestinal bleeding similarly supports these protective effects. Kato et al. [46] found that the combination of PPI and rebamipide is more effective than PPI alone for treating ulcers after endoscopic submucosal dissection. Mizukami et al. [47] reported that rebamipide is effective in combination therapy with PPIs to prevent low-dose aspirin-induced gastrointestinal symptoms. Therefore, co-prescription of rebamipide with PPIs may be effective in specific patients.

Our study found that >80% of NSAID users with orthopaedic conditions were older than 65 years of age. Maes et al. [14] showed that long-term PPI use in patients older than 60 years increased the incidence of fractures due to osteoporosis. Zhang et al. [48] reported PPI use may influence total hip bone mineral density via plasma metabolites involved in the sex hormone pathway. The FDA has stated that low-dose, short-term oral PPIs are not associated with fractures; moreover, fracture warnings owing to high-dose, long-term PPIs are still present in the FDA's current safety announcement [49]. Yamashiro et al. [50] reported PPI use is associated with an increased risk of hypomagnesemia, especially in males under 60 years and those with low body-mass index. Several systematic reviews suggest that suppressing gastric acid with PPI leads to bacterial colonization and increased susceptibility to enteric infections [51, 52]. A case-cohort study by Bruin et al. [16] found that perioperative dosing of PPIs is associated with increased prosthetic joint infections. According to a multicenter, cross-sectional study

**Table 2. Conditional logistic regression analysis of upper gastrointestinal bleeding.**

| Prescription status of rebamipide | Case (*n* = 215) | Control (*n* = 1,516) | Crude Odds Ratio (95% CI) | Adjusted Odds Ratio[a] (95% CI) |
|---|---|---|---|---|
| Non-user | 93 (43.3) | 724 (47.8) | 1 (Reference) | 1 (Reference) |
| Continuous-user | 52 (24.2) | 597 (39.4) | 0.68 (0.47–0.97) | 0.65 (0.44–0.96) |
| Irregular-user | 70 (32.6) | 195 (12.9) | 2.79 (1.97–3.95) | 2.57 (1.73–3.81) |

[a]Conditional logistic regression analysis was conducted by considering matching factors: (A) sex at $t_0$, (B) age (± 5 years) at $t_0$, (C) follow-up (± 180 days) from $t_0$ to index date, (D) the total number of NSAID tablets prescribed, and (E) the total dose of the five types of NSAIDs (loxoprofen, celecoxib, diclofenac, meloxicam, and ibuprofen).

CI, confidence interval; NSAIDs, non-steroidal anti-inflammatory drugs.

conducted in 2022 in Japan, the mean age of new patients with hip osteoarthritis was 63.5 years [53], and those patients with osteoarthritis may undergo joint replacement surgery after taking NSAIDs for an extended duration [54]. However, infection and periprosthetic fractures are serious complications of joint replacement surgery [55]. Using PPIs to prevent NSAID-induced ulcers in patients at low risk should be discussed in light of these possibilities.

The CYP2C19 enzyme involved in PPI metabolism varies considerably across different racial and ethnic groups [18, 56]. Asians with low CYP2C19 levels and lower stomach acid secretion [57] than Caucasians could have an increased risk of adverse drug reactions associated with PPIs [18]. A recent study by Lee et al. using the Korean database found that PPIs were associated with severe COVID-19-related outcomes [17]. However, Nayar et al. [19] found no such association among United States veterans. In addition to the effects of unadjusted confounding factors [20], racial differences, such as Asians having an increased likelihood of experiencing adverse drug reactions from PPIs, could be related to the discrepancy in results.

Forgerini et al. [58] reported variants in the *CYP2C9* gene, as the *2 and *3 alleles are responsible for the slow metabolism of NSAIDs, which increases the risk of gastrointestinal bleeding. Zhou et al. [59] reported pronounced regional differences in the distribution of the *CYP2C9*2 and *3 alleles, underscoring the importance of genetic variability in the *CYP2C9* gene for the efficacy and safety of drugs on a global scale. When considering the risk of NSAID-induced ulcers in the future, it is essential to consider such genetic backgrounds to enhance patient safety and tailor treatment strategies.

This study has several limitations. First, the cases in this study underwent upper gastrointestinal endoscopy and had a diagnosis of upper gastrointestinal bleeding based on ICD-10 codes. However, we did not have access to the written reports or images of the endoscopy; thus, there might have been some misclassification of cases in this study. Second, we did not assume that upper gastrointestinal bleeding was due to etiologies other than NSAID-induced ulcers. Therefore, we ensured the robustness of our findings by conducting sensitivity analyses that excluded patients with a Charlson Comorbidity Index score of 1 or higher to support the assumption that the upper gastrointestinal bleeding was not due to other causes, such as neoplasia, cirrhosis, or varicose veins. Third, this study did not adjust for confounding factors that cannot be captured by medical receipt data, such as patients' smoking and alcohol consumption habits.

The strength of this study is in its high tracking rate. Japan has a universal health insurance system, in which most citizens visit hospitals and receive prescriptions for NSAIDs and gastroprotective medicines from their doctors. Since all individuals in this study visited a hospital and are covered by insurance, paying a maximum of 30% out-of-pocket for prescription medication, we assume that there is a low likelihood they would purchase NSAIDs at significantly higher prices by not using their insurance. Our results using medical claims data might reflect the incidence and risk across the broader population.

## Conclusions

Our study showed that continuous co-prescription of rebamipide significantly reduced the risk of upper gastrointestinal bleeding in new users of NSAIDs with osteoarthritis or low back pain without risk factors other than age. This result was confirmed by a nested case-control study, which could strictly adjust the type and dosage of NSAIDs prescribed.

## Supporting information

**S1 Fig. Days from $t_0$ to upper gastrointestinal bleeding.**
(TIF)

**S1 Table. ATC code list.**
(DOCX)

**S2 Table. Demographics of the cohort in which NSAIDs were first used for osteoarthritis or back pain.**
(DOCX)

**S3 Table. Population that received NSAIDs for the first time for osteoarthritis or low back pain and without risk factors for gastric ulcer other than age.**
(DOCX)

**S4 Table. Conditional logistic regression analysis of upper gastrointestinal bleeding in patients aged under 65 years (*Subgroup analysis 1*).**
(DOCX)

**S5 Table. Conditional logistic regression analysis of upper gastrointestinal bleeding in patients aged 65 years or over (*Subgroup analysis 2*).**
(DOCX)

**S6 Table. Conditional logistic regression analysis of upper gastrointestinal bleeding in patients receiving loxoprofen at $t_0$ (*Subgroup analysis 3*).**
(DOCX)

**S7 Table. Conditional logistic regression analysis of upper gastrointestinal bleeding in patients receiving celecoxib at $t_0$ (*Subgroup analysis 4*).**
(DOCX)

**S8 Table. Conditional logistic regression analysis of upper gastrointestinal bleeding (*Sensitivity analysis 1*).**
(DOCX)

**S9 Table. Conditional logistic regression analysis of upper gastrointestinal bleeding (*Sensitivity analysis 2*).**
(DOCX)

**S10 Table. Conditional logistic regression analysis of upper gastrointestinal bleeding (*Sensitivity analysis 3*).**
(DOCX)

## Author Contributions

**Conceptualization:** Satoshi Yamate, Haruhisa Fukuda, Yasuharu Nakashima.

**Data curation:** Satoshi Yamate, Haruhisa Fukuda.

**Formal analysis:** Satoshi Yamate.

**Funding acquisition:** Haruhisa Fukuda.

**Investigation:** Satoshi Yamate, Chieko Ishiguro, Haruhisa Fukuda.

**Methodology:** Satoshi Yamate, Chieko Ishiguro, Satoshi Hamai.

**Project administration:** Haruhisa Fukuda, Yasuharu Nakashima.

**Resources:** Haruhisa Fukuda.

**Software:** Satoshi Yamate.

**Supervision:** Chieko Ishiguro, Haruhisa Fukuda, Satoshi Hamai, Yasuharu Nakashima.

**Validation:** Haruhisa Fukuda.

**Visualization:** Satoshi Yamate.

**Writing – original draft:** Satoshi Yamate.

**Writing – review & editing:** Satoshi Yamate, Chieko Ishiguro, Haruhisa Fukuda, Satoshi Hamai, Yasuharu Nakashima.

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
