## [Decision Letter · Decision Letter 0]

23 Jan 2024

PONE-D-23-40113Rebamipide prevents upper gastrointestinal bleeding in NSAID use for orthopaedic conditions: A nested case-control study using the LIFE Study databasePLOS ONE

Dear Dr. Hamai,

Thank you for submitting your manuscript to PLOS ONE. After careful consideration, we feel that it has merit but does not fully meet PLOS ONE’s publication criteria as it currently stands. Therefore, we invite you to submit a revised version of the manuscript that addresses the points raised during the review process.

**ACADEMIC EDITOR:**

Thank you for submitting your manuscript, titled "[Rebamipide prevents upper gastrointestinal bleeding in NSAID use for orthopaedic conditions: A nested case-control study using the LIFE Study database]," to PLOS ONE. Your work has now been reviewed by two experts in the field, and I have carefully considered their evaluations.

After a thorough assessment, I have decided that your manuscript could potentially be suitable for publication, but it requires major revisions. Both reviewers have acknowledged the importance and relevance of your study but have raised significant concerns that need to be addressed to meet our publication standards.

We encourage you to resubmit your revised manuscript. Please include a detailed response letter outlining how you have addressed each of the reviewers' comments.

Best regards

We look forward to receiving your revised manuscript.

Kind regards,

Mahmoud Kandeel

Academic Editor

PLOS ONE

“This work was supported by the Japan Society for the Promotion of Science KAKENHI (Grant Numbers JP19K21590 and JP20H00563) and by Japan Science and Technology Agency FOREST (Fusion Oriented REsearch for disruptive Science and Technology) program (Grant Number JPMJFR205J).”

“We would like to thank Editage (www.editage.com) for English language editing (funding source: JPMJFR205J).”

“This work was supported by the Japan Society for the Promotion of Science KAKENHI (Grant Numbers JP19K21590 and JP20H00563) and by Japan Science and Technology Agency FOREST (Fusion Oriented REsearch for disruptive Science and Technology) program (Grant Number JPMJFR205J).”

“Authors with competing interests

I have read the journal's policy and the authors of this manuscript have the following competing interests:Yasuharu Nakashima received research grants from Chugai Pharmaceutical Co., Ltd., AYUMI Pharmaceutical Corporation, EA Pharma Co., Ltd., KYOCERA Corporation, and Zimmer Biomet G.K. All other authors have no conflicts of interest to disclose.”

6. We note that you have indicated that there are restrictions to data sharing for this study. PLOS only allows data to be available upon request if there are legal or ethical restrictions on sharing data publicly. For more information on unacceptable data access restrictions, please see http://journals.plos.org/plosone/s/data-availability#loc-unacceptable-data-access-restrictions.

Reviewers' comments:

Reviewer's Responses to Questions

**Comments to the Author**

1. Is the manuscript technically sound, and do the data support the conclusions?

Reviewer #1: Yes

Reviewer #2: Partly

2. Has the statistical analysis been performed appropriately and rigorously? 

Reviewer #1: Yes

Reviewer #2: Yes

3. Have the authors made all data underlying the findings in their manuscript fully available?

Reviewer #1: Yes

Reviewer #2: No

4. Is the manuscript presented in an intelligible fashion and written in standard English?

Reviewer #1: Yes

Reviewer #2: Yes

5. Review Comments to the Author

Reviewer #1: Estimated Editor and Authors,

First, I would like to thank you for the opportunity to review this manuscript. This is a nested case-control that evaluated the effectiveness of the prescription of rebamipide for prevention of upper gastrointestinal bleeding in NSAIDs users living with orthopedic conditions. The manuscript is well written and is relevant in the scope of patient safety, considering the high morbidity and mortality related to gastrointestinal bleeding as an adverse drug event. However, methodological reporting is somewhat confusing and needs to be improved for a better understanding of the study and to be reproducible.

Follow my feedback below to help you improve the quality and clarity of your manuscript:

- Title: The authors report as if the rebamipide prevents upper gastrointestinal bleeding on any NSAIDS user, but the authors' findings only identified this protection in occasional users. Therefore, it is important to review the title so that it is harmonious with the findings of the study.

- At many times the occasional use of NSAIDS is cited, but this is not defined. It is very important to define how stratified the consumption of NSAIDS and what was considered in each stratum.

Introduction

- Page 5, Line 60: Is there no more current data on the rebamipide prescription?

- The most appropriate terminology is "adverse drug events" and not "effects".

- Important to define what was considered "low risk of NSAIDs induced ulcers".

- Page 6, lines 86 and 87 do not fit in the introduction: "We used medical claims data for the analysis we consider it a useful approach for tracking cats across specialties in the real world."

Methodology

- The methodology should follow the reporting of checklist Strobe, because in many moments this section is confused. Strobe should be used to guide the methodology report.

I understand, for example, that the methodology should start by presenting the study and setting design and not the database used to consult the data.

- Page 6, Line 103: Do two public insurance systems cover the entire country?

- How many researchers conducted data collection? Were pre-delineated forms used for this data collection?

- Why was the Time Window (T.W.) from 90 days before t0 to 1 day before t0 the exclusion criteria?

- Were the life habits of patients, such as smoking and consumption of beverages, which depending on the amount consumed may be associated with gastrointestinal problems?

- Pages 8 and 9, lines 151 - 153: This sentence is not clear. What refers to censorship in line 153?

- The definition of cases is not clear. Patients diagnosed with osteoarthritis or back pain who were prescribed from NSAIDs were follow-up until these patients undergoing endoscopy and developed upper gastrointestinal bleeding? This flow of recruitment of participants was not clear to me.

- If the patient had in his medical record the reporting of endoscopic findings such as "duodenal ulcer" or "gastric ulcer", would it be considered case? As already reported in the limitations of the study, the non-analysis of endoscopic reports consists of an important bias.

Another aspect, the intention was to evaluate upper gastrointestinal bleeding as an adverse event of the use of NSAIDS, right? Therefore, this excludes the non-variceal etiology. How was the control of the etiologies of the upper gastrointestinal bleeding? Was it possible to identify if the patient was bleeding by neoplasia, cirrhosis, varicose veins, or other conditions? This aspect needs to be better reported.

- I was also in doubt in the following excerpt "However, we assume that the fictitious disease name of "upper gastrointestinal bleeding" would be rarely assigned to a patent without any symptoms of gastrointestinal ulcer." Was it considered cases only patients with upper with gastrointestinal bleeding or just with ulcer as well? Were evaluated if these patients had signs of recent bleeding?

- Page 10, line 176: define what was considered "1) continuous user, 2) occasional user, and 3) non-user".

- Page 11, line 206: the following sentence is confusing: "the analytical results are planned to be disseminated to the public through these participating municipalities."

- One piece of information that would greatly enrich the results would be to assess the defined daily dose of NSAIDs consumed by the participants. I suggest that the authors assess the feasibility or relevance of including this data.

- Was it possible to control whether patients self-medicated with NSAIDs, in addition to the medical prescription they received? How are NSAIDs sold in Japan? In many patients, there is free access to this class.

Results

- When reporting averages or medians, I suggest including the standard deviation or quartiles, respectively.

- In table 1, the data presented in the last rows is not clear. Is it the dose of each NSAID and the respective quartiles? Why are most of the drugs set to 0? I think this form of presentation is a bit confusing. I apologize if I'm not understanding the presentation correctly.

Discussion

- As the authors cite the importance of variants in the CYP2C19 gene, which can impact the metabolism of proton pump inhibitors, it is also worth highlighting variants in the CYP2C9 gene, as the *2 and *3 alleles are responsible for the slow metabolism of NSAIDs, which increases the risk of gastrointestinal bleeding.

- The authors could further highlight the clinical implications and contributions of these findings to clinical practice and patient safety.

Reviewer #2: This is a worthwhile study as there is a paucity of clinical trial data of rebamipide for this indication.

Methods are sound. Inclusion criteria are quite narrow but this helps to avoid confusing.

The discussion about the adverse effects of PPIs compared to those of rebamipide is not equitable. More types of adverse effects have been reported with PPIs, but they have also been used more widely than rebamipide.

It is necessary to make more mention of the adverse effects of rebapimide. The labels list leukopenia and liver dysfunction as adverse effects; although rare (I checked Malaysian and Philippine labels).

The results are than continuous use of rebamipide was associated with less GI bleeding than no use, but the occasional use was associated with a higher risk than no use

The conclusion than rebamipide may be more beneficial than PPIs is unsound since the effect of PPIs has not be addressed in this study.

Some issues need to be addressed:

In the abstract, it should be specified that the patients are at low risk of NSAIDs induced ulcers.

Enteric infections and hypomagnesemia should be mentioned as adverse effects of PPI

Line 78: “[Rebamipide] and may be superior to PPIs as a prophylactic agent for patients at low risk of NSAID induced ulcers.” This statement is not supported by a reference.

Line 80 “The prevention of NSAID-induced ulcers can be achieved from the initial prescription of NSAIDs, such as in an orthopaedic outpatient clinic; however…”

This phrase is strange. From the initial prescription of NSAIDs, prevention is established, it is not yet achieved.

The methods should make it clear whether it is a calendar month or a 30-day period.

Why is ICD-10 code B98.0 (H. pylori as a cause of diseases classified in other chapters) not used as an exclusion criterion?

Line 138: Anticoagulants are associated with bleeding rather than ulcer risk.

Authors should provide unadjusted estimates of the main analysis. (STROBE item 16)

Line 162: “However, we assumed that the fictitious disease name…”

The phrase is strange. The name is not fictitious.

Line 170: the total number of NSAID tablets “prescribed” instead of “administered”.

Line 320: “the study sample was limited to low-risk patients”. This is not a limitation since the research question was about low risk patients

S2 Figure The duration of upper gastrointestinal bleeding after the first prescription of NSAIDs

Figure title is wrong. Data are days to bleeding.

6. PLOS authors have the option to publish the peer review history of their article (what does this mean?). If published, this will include your full peer review and any attached files.

Reviewer #1: **Yes: **Marcela Forgerini

Reviewer #2: **Yes: **Javier Garjon

---

## [Author Response · Author response to Decision Letter 0]

11 Mar 2024

Response: Thanks for the information.

We have revised the manuscript according to the PLOS ONE's style requirements.

 Response: We reconfirm that we had clearly stated in the main text as follows.

 Text Changes (Materials and methods, Lines 120-122):

The review board waived the requirement for informed consent due to the study’s retrospective nature and because all records were de-identified and fully anonymized before our access for analysis.

“This work was supported by the Japan Society for the Promotion of Science KAKENHI (Grant Numbers JP19K21590 and JP20H00563) and by Japan Science and Technology Agency FOREST (Fusion Oriented REsearch for disruptive Science and Technology) program (Grant Number JPMJFR205J).”

Response: Please add the following statement to our section of our “Funding”:

 Text Changes (Funding):

This work was supported by the Japan Society for the Promotion of Science KAKENHI (Grant Numbers JP19K21590 and JP20H00563) and by Japan Science and Technology Agency FOREST (Fusion Oriented REsearch for disruptive Science and Technology) program (Grant Number JPMJFR205J). The funders had no role in study design, data collection and analysis, decision to publish, or preparation of the manuscript.

“We would like to thank Editage (www.editage.com) for English language editing (funding source: JPMJFR205J).”

“This work was supported by the Japan Society for the Promotion of Science KAKENHI (Grant Numbers JP19K21590 and JP20H00563) and by Japan Science and Technology Agency FOREST (Fusion Oriented REsearch for disruptive Science and Technology) program (Grant Number JPMJFR205J).”

Response: We removed the funding-related text from the manuscript.

Please revise our “Funding” section as follows.

Text Changes (Funding):

This work was supported by the Japan Society for the Promotion of Science KAKENHI (Grant Numbers JP19K21590 and JP20H00563) and by Japan Science and Technology Agency FOREST (Fusion Oriented REsearch for disruptive Science and Technology) program (Grant Number JPMJFR205J). The funders had no role in study design, data collection and analysis, decision to publish, or preparation of the manuscript.

“Authors with competing interests

I have read the journal's policy and the authors of this manuscript have the following competing interests:Yasuharu Nakashima received research grants from Chugai Pharmaceutical Co., Ltd., AYUMI Pharmaceutical Corporation, EA Pharma Co., Ltd., KYOCERA Corporation, and Zimmer Biomet G.K. All other authors have no conflicts of interest to disclose.”

Response: Please add the following statement to our “Competing Interests” section:

"This does not alter our adherence to PLOS ONE policies on sharing data and materials."

 Text Changes (Competing Interests):

Yasuharu Nakashima received research grants from Chugai Pharmaceutical Co., Ltd., AYUMI Pharmaceutical Corporation, EA Pharma Co., Ltd., KYOCERA Corporation, and Zimmer Biomet G.K. All other authors have no conflicts of interest to disclose. This does not alter our adherence to PLOS ONE policies on sharing data and materials.

6. We note that you have indicated that there are restrictions to data sharing for this study. PLOS only allows data to be available upon request if there are legal or ethical restrictions on sharing data publicly. For more information on unacceptable data access restrictions, please see http://journals.plos.org/plosone/s/data-availability#loc-unacceptable-data-access-restrictions.

 Response: Please revise our Data Availability Statement as follows.

Text Change (Data Availability Statement):

Data Availability Statement: The data used in this study were acquired under agreements between Kyushu University and the participating municipalities, which stipulate that the data can only be used by authorized research institutions and cannot be shared with third parties. However, research institutions that have entered into agreements with the authorized research group in Kyushu University may access the data. Please contact the Joint Research Department of Kyushu University (ijkkyoudou@jimu.kyushu-u.ac.jp) regarding data access.  

Response to Reviewer #1:

Reviewer #1 (1): Estimated Editor and Authors,

First, I would like to thank you for the opportunity to review this manuscript. This is a nested case-control that evaluated the effectiveness of the prescription of rebamipide for prevention of upper gastrointestinal bleeding in NSAIDs users living with orthopedic conditions. The manuscript is well written and is relevant in the scope of patient safety, considering the high morbidity and mortality related to gastrointestinal bleeding as an adverse drug event. However, methodological reporting is somewhat confusing and needs to be improved for a better understanding of the study and to be reproducible.

Follow my feedback below to help you improve the quality and clarity of your manuscript:

Response: We are grateful to Reviewer #1 for critical comments and useful suggestions that helped us to improve our paper considerably. We appreciate you pointing out that the description of our methodology was unclear. We did our best effort to revise the manuscript according to your comments.

Reviewer #1 (2):

- Title: The authors report as if the rebamipide prevents upper gastrointestinal bleeding on any NSAIDS user, but the authors' findings only identified this protection in occasional users. Therefore, it is important to review the title so that it is harmonious with the findings of the study.

 Response: 

Thank you for your valuable comment. We identified this protection in continuous users. We have revised the title to be appropriate.

 Text Changes (Title):

Continuous co-prescription of rebamipide prevents upper gastrointestinal bleeding in NSAID use for orthopaedic conditions: A nested case-control study using the LIFE Study database

Reviewer #1 (3):

- At many times the occasional use of NSAIDS is cited, but this is not defined. It is very important to define how stratified the consumption of NSAIDS and what was considered in each stratum.

Response: We have modified the wording as we believe Irregular-user would be more appropriate than occasional-user. We also added a definition of grouping to Abstract. In addition, we added a figure to the main text that illustrates the definition of the grouping.

 Text Changes (Abstract, Lines 36-41):

Exposure to rebamipide was defined as prescription status from t0 to index date: Non-user (rebamipide was not co-prescribed during the follow-up period), Continuous-user (rebamipide was co-prescribed from t0 with the same number of tablets as NSAIDs), and Irregular-user (neither Non-user nor Continuous-user). Conditional logistic regression analysis was conducted to estimate each category’s odds ratio compared to non-users.

 Text Changes (Materials and methods, Lines 177-182):

Patients were grouped based on the prescription status of rebamipide from t0 to the index date: 1) Non-user, 2) Continuous-user, and 3) Irregular-user (Fig 1). Patients without rebamipide prescriptions from t0 to the index date were grouped as non-users. If patients were continuously prescribed rebamipide from t0 to the index date with the same number of tablets as NSAIDs, they were grouped as continuous-users. Patients not falling under either continuous-user or non-user were grouped as irregular-users.

 Text Changes (Fig 1., Lines 184-188):

Fig 1. Definition of the prescription status of rebamipide. Non-user = Rebamipide was not co-prescribed during the follow-up period. Continuous-user = Rebamipide was co-prescribed from t0 with the same number of tablets as NSAIDs. Irregular-user = Neither Non-user nor Continuous-user. NSAIDs, non-steroidal anti-inflammatory drugs.

Reviewer #1 (4):

Introduction

- Page 5, Line 60: Is there no more current data on the rebamipide prescription?

Response:

We thank the reviewer for the valuable comment.

We have checked the latest data available and updated the numbers.

 Text Changes (Introduction, Lines 51-53):

According to the government’s annual all-counts survey, 2.5 billion tablets (0.25 million kg) of rebamipide, marketed by 24 pharmaceutical companies, were prescribed from April 2021 to March 2022.

Reviewer #1 (5):

- The most appropriate terminology is "adverse drug events" and not "effects".

Response: We thank the reviewer for the valuable comment. We agree that the term "adverse drug events" was appropriate and have revised the relevant section.

 Text Changes (Introduction, Lines 68-69): 

Recently, there have been several reports linking PPIs with adverse drug events,

Text Changes (Introduction, Lines 72-73):

On the other hand, rebamipide prevents ulcers without affecting gastric acid secretion[1], is known to have very few adverse drug events,

Text Changes (Introduction, Lines 284-285):

They also found that rebamipide was associated with fewer adverse drug events,

Text Changes (Discussions, Lines 326-327):

than Caucasians could have an increased risk of adverse drug events associated with PPIs.

Text Changes (Discussions, Lines 330-331):

such as Asians having an increased likelihood of experiencing adverse drug events from PPIs,

Reviewer #1 (6):

- Important to define what was considered "low risk of NSAIDs induced ulcers".

Response: We thank the reviewer for the valuable comment. We listed the risk factors and clarified that patients without these factors were defined as low risk patients.

Text Changes (Introduction, Lines 62-68):

The effectiveness of PPIs in preventing NSAID-induced ulcers has been reported in numerous randomized controlled trials and is also recommended by worldwide guidelines for moderate or high-risk patients who have any of the following risk factors: age over 65 years, previous history of an uncomplicated ulcer, or concurrent use of aspirin, antiplatelet drugs, corticosteroids, or anticoagulant agents. However, their effectiveness in low-risk patients without risk factors remains unclear.

Reviewer #1 (7):

- Page 6, lines 86 and 87 do not fit in the introduction: "We used medical claims data for the analysis we consider it a useful approach for tracking cats across specialties in the real world."

Response: We thank the reviewer for the valuable comment. We have moved the relevant sentence from Introduction to Methods.

Text Changes (Materials and methods, Lines 103-104):

We used medical claims data for the analysis as we considered it a useful approach for tracking patients across specialties in the real world.

Reviewer #1 (8):

Methodology

- The methodology should follow the reporting of checklist Strobe, because in many moments this section is confused. Strobe should be used to guide the methodology report.

I understand, for example, that the methodology should start by presenting the study and setting design and not the database used to consult the data.

Response: We have rearranged the method headings and the order of content to follow STROBE whenever possible.

 Text Changes (Materials and methods, Lines 88-100):

Study design

 Text Changes (Materials and methods, Lines 102-124):

 Setting

Text Changes (Materials and methods, Lines 126-187):

 Participants

Reviewer #1 (9):

- Page 6, Line 103: Do two public insurance systems cover the entire country?

Response: The two public insurance programs cover the majority of the older population, however, do not cover the entire population. Therefore, we added a sentence that specifically indicates the coverage rate.

Text Changes (Materials and methods, Lines 113-116):

The forme

---

## [Decision Letter · Decision Letter 1]

7 May 2024

PONE-D-23-40113R1Continuous co-prescription of rebamipide prevents upper gastrointestinal bleeding in NSAID use for orthopaedic conditions: A nested case-control study using the LIFE Study databasePLOS ONE

Dear Dr. Hamai,

Thank you for submitting your manuscript to PLOS ONE. After careful consideration, we feel that it has merit but does not fully meet PLOS ONE’s publication criteria as it currently stands. Therefore, we invite you to submit a revised version of the manuscript that addresses the points raised during the review process.

We look forward to receiving your revised manuscript.

Kind regards,

Mahmoud Kandeel

Academic Editor

PLOS ONE

Journal Requirements:

Reviewers' comments:

Reviewer's Responses to Questions

**Comments to the Author**

1. If the authors have adequately addressed your comments raised in a previous round of review and you feel that this manuscript is now acceptable for publication, you may indicate that here to bypass the “Comments to the Author” section, enter your conflict of interest statement in the “Confidential to Editor” section, and submit your "Accept" recommendation.

Reviewer #1: All comments have been addressed

Reviewer #2: (No Response)

2. Is the manuscript technically sound, and do the data support the conclusions?

Reviewer #1: Yes

Reviewer #2: Partly

3. Has the statistical analysis been performed appropriately and rigorously? 

Reviewer #1: Yes

Reviewer #2: Yes

4. Have the authors made all data underlying the findings in their manuscript fully available?

Reviewer #1: Yes

Reviewer #2: No

5. Is the manuscript presented in an intelligible fashion and written in standard English?

Reviewer #1: Yes

Reviewer #2: Yes

6. Review Comments to the Author

Reviewer #1: Dear Editor and Authors,

Firstly, I thank you for the opportunity to review again this manuscript.

The authors carried out a carefully review and attended all my suggestions. Hence, I am delighted with the revised version and have no other suggestions.

Minor comment: genes should be written in italics.

Finally, I congratulate the authors for their careful review.

Reviewer #2: Continuous-user is defined as one whom rebamipide was co-prescribed from t0 with the same number of tablets as NSAIDs. However, as dosage of rebamipide is one tablet 3 times/day and dosage of celecoxib is one tablet/day, the same number of tablets covers fewer days with rebamipide than with celecoxib. This not fit with figure 1.

In which group is someone who is prescribed more rebamipide tablets than NSAIDs (as they should be in the case of celecoxib or meloxicam for being a continuous user)? It is necessary to clarify this issue.

I am sorry, but I disagree with reviewer #1. “Adverse drug event” is not a standard term; I consider “adverse effect” or “adverse drug reaction” more accurate terms because a causal relationship between the drug and an occurrence is suspected.

Please see: European Medicines Agency and Heads of Medicines Agencies, 2017. Guideline on good pharmacovigilance practices (GVP) Annex I - Definitions (Rev 4). https://www.ema.europa.eu/en/documents/scientific-guideline/guideline-good-pharmacovigilance-practices-annex-i-definitions-rev-4_en.pdf

Lines 292-296

There is a duplicity:

prescribers should be aware of the risk of pulmonary adverse drug events associated with rebamipide [44], and leukopenia might occur in rare cases [5], meaning that rebamipide is not entirely risk-free. However, prescribers need to be aware of the potential for pulmonary adverse events associated with rebamipide [44] and rare instances of leukopenia [5], highlighting that it is not without risks.

Line 323

"Using PPIs to prevent NSAID-induced ulcers in patients with no risks should be discussed"

“with no risk factors” or “at low risk” are better expressions.

Lines 345 346

"While we found that NSAIDs might unexpectedly cause upper gastrointestinal bleeding, this conclusion is based on studies with low-risk patients."

I do not understand the sentence. NSAIDs, expectedly, cause upper gastrointestinal bleeding even in low-risk patients. This is a finding from the present study. What studies are the authors referring to?

7. PLOS authors have the option to publish the peer review history of their article (what does this mean?). If published, this will include your full peer review and any attached files.

Reviewer #1: **Yes: **Marcela Forgerini

Reviewer #2: No

---

## [Author Response · Author response to Decision Letter 1]

17 May 2024

Response to Journal requirements:

Response:

Thanks for the information. We re-searched all the articles on Pubmed and confirmed that they had not been Retracted.

We have added one additional citation (Sato T, Yamate S, Utsunomiya T, Inaba U, Ike H, Kinoshita K, et al. Life Course Epidemiology of Hip Osteoarthritis in Japan: A Multicenter, Cross-Sectional Study. J Bone Joint Surg Am. 2024; Forthcoming. doi: 10.2106/JBJS.23.01044, PMID 38626018.) and have indicated accordingly at the end of this letter.

Response to Reviewer #1:

Reviewer #1: Dear Editor and Authors,

Firstly, I thank you for the opportunity to review again this manuscript.

The authors carried out a carefully review and attended all my suggestions. Hence, I am delighted with the revised version and have no other suggestions.

Minor comment: genes should be written in italics.

Finally, I congratulate the authors for their careful review.

Response:

We are grateful to Reviewer #1 for taking the time to review and provide comments to improve the quality of our paper. As commented to us, we have modified the genes to italics.

Text Changes (Discussion, Lines 338-345):

Forgerini et al. reported variants in the CYP2C9 gene, as the *2 and *3 alleles are responsible for the slow metabolism of NSAIDs, which increases the risk of gastrointestinal bleeding. Zhou et al. reported pronounced regional differences in the distribution of the CYP2C9*2 and *3 alleles, underscoring the importance of genetic variability in the CYP2C9 gene for the efficacy and safety of drugs on a global scale. When considering the risk of NSAID-induced ulcers in the future, it is essential to consider such genetic backgrounds to enhance patient safety and tailor treatment strategies.

Response to Reviewer #2:

Reviewer #2 (1):

Continuous-user is defined as one whom rebamipide was co-prescribed from t0 with the same number of tablets as NSAIDs. However, as dosage of rebamipide is one tablet 3 times/day and dosage of celecoxib is one tablet/day, the same number of tablets covers fewer days with rebamipide than with celecoxib. This not fit with figure 1.

In which group is someone who is prescribed more rebamipide tablets than NSAIDs (as they should be in the case of celecoxib or meloxicam for being a continuous user)? It is necessary to clarify this issue.

Response:

Thank you for your comment. You are correct in noting that rebamipide is recommended to be taken three times per day, while celecoxib is usually taken twice per day in Japan. We believe that the clinical significance of our results lies in showing that a prescribing pattern where rebamipide and NSAIDs taken with the same number of tablets can be effective. This approach may enhance patient compliance in real-world settings because complex prescribing patterns may reduce patient compliance in real-world clinical practice. We have revised the manuscript to clarify this issue. Thank you for your constructive feedback.

Text Changes (Materials and methods, Lines 182-185):

Therefore, if a patient was prescribed NSAIDs less than three times a day but rebamipide three times a day [4], they were classified as an irregular user because the number of tablets did not match.

Text Changes (Discussion, Lines 275-277):

Using rebamipide for preventing NSAID-induced ulcers, a simple co-prescription pattern for rebamipide may improve adherence in clinical practice.

Reviewer #2 (2):

I am sorry, but I disagree with reviewer #1. “Adverse drug event” is not a standard term; I consider “adverse effect” or “adverse drug reaction” more accurate terms because a causal relationship between the drug and an occurrence is suspected.

Please see: European Medicines Agency and Heads of Medicines Agencies, 2017. Guideline on good pharmacovigilance practices (GVP) Annex I - Definitions (Rev 4). https://www.ema.europa.eu/en/documents/scientific-guideline/guideline-good-pharmacovigilance-practices-annex-i-definitions-rev-4_en.pdf

Response:

Thank you for the information. We have confirmed the PDFs and corrected the terminology to “adverse drug reaction”. Thank you for your careful review to improve the quality of our paper.

Text Changes (Introduction, Lines 68-71):

Recently, there have been several reports linking PPIs with adverse drug reactions, such as dementia, bone fractures, prosthetic joint infection, and severe clinical outcomes of coronavirus disease (COVID-19).

Text Changes (Introduction, Lines 72-75):

On the other hand, rebamipide prevents ulcers without affecting gastric acid secretion, is known to have very few adverse drug reactions, and could be considered as an alternative to PPIs as a prophylactic agent for patients at low risk of NSAID-induced ulcers.

Text Changes (Discussion, Lines 287-299):

Rebamipide was comparable to misoprostol in reducing the incidence of gastric and duodenal ulcers according to a multicenter, double-blind, randomized parallel-group study by Kim et al. They also found that rebamipide was associated with fewer adverse drug reactions, such as nausea and diarrhea. However, our study found that the actual number of prescriptions is approximately 1/100th of that for rebamipide; this could be because rebamipide has fewer adverse drug reactions than misoprostol. Recently, Lee et al. reported no significant difference in the risk of serious gastrointestinal adverse drug reactions between PPI and rebamipide for preventing NSAID-induced ulcers (adjusted hazard ratio 0.69, 95% CI 0.27–1.76) using the national claims database. Even though the frequency is lower compared to other acid-related disorder treatments, prescribers should be aware of the risk of pulmonary adverse drug reactions associated with rebamipide, and leukopenia might occur in rare cases, meaning that rebamipide is not entirely risk-free.

Text Changes (Discussion, Lines 330-332):

Asians with low CYP2C19 levels and lower stomach acid secretion than Caucasians could have an increased risk of adverse drug reactions associated with PPIs.

Text Changes (Discussion, Lines 334-337):

In addition to the effects of unadjusted confounding factors, racial differences, such as Asians having an increased likelihood of experiencing adverse drug reactions from PPIs, could be related to the discrepancy in results.

Reviewer #2 (3):

Lines 292-296

There is a duplicity:

prescribers should be aware of the risk of pulmonary adverse drug events associated with rebamipide [44], and leukopenia might occur in rare cases [5], meaning that rebamipide is not entirely risk-free. However, prescribers need to be aware of the potential for pulmonary adverse events associated with rebamipide [44] and rare instances of leukopenia [5], highlighting that it is not without risks.

Response:

We are grateful for the careful review of Reviewer #2. We have removed the duplicate sections of the latter.

Text Changes (Discussion, Lines 296-299):

Even though the frequency is lower compared to other acid-related disorder treatments, prescribers should be aware of the risk of pulmonary adverse drug reactions associated with rebamipide [44], and leukopenia might occur in rare cases [5], meaning that rebamipide is not entirely risk-free. However, prescribers need to be aware of the potential for pulmonary adverse events associated with rebamipide [44] and rare instances of leukopenia [5], highlighting that it is not without risks. 

Reviewer #2 (4):

Line 323

"Using PPIs to prevent NSAID-induced ulcers in patients with no risks should be discussed"

"with no risk factors" or "at low risk" are better expressions.

Response:

 Thank you for your comment. We have revised our expressions.

Text Changes (Discussion, Lines 326-328):

Using PPIs to prevent NSAID-induced ulcers in patients at low risk should be discussed in light of these possibilities.

Reviewer #2 (5):

Lines 345 346

"While we found that NSAIDs might unexpectedly cause upper gastrointestinal bleeding, this conclusion is based on studies with low-risk patients."

I do not understand the sentence. NSAIDs, expectedly, cause upper gastrointestinal bleeding even in low-risk patients. This is a finding from the present study. What studies are the authors referring to?

Response: Thank you for your comment. In response to the previous comment by Reviewer #1, our intention was not well expressed.

(Previous comment by Reviewer #1: Another aspect, the intention was to evaluate upper gastrointestinal bleeding as an adverse event of the use of NSAIDS, right? Therefore, this excludes the non-variceal etiology. How was the control of the etiologies of the upper gastrointestinal bleeding? Was it possible to identify if the patient was bleeding by neoplasia, cirrhosis, varicose veins, or other conditions? This aspect needs to be better reported.)

Text Changes (Materials and methods, Lines 350-354):

Second, we did not assume that upper gastrointestinal bleeding was due to etiologies other than NSAID-induced ulcers. Therefore, we ensured the robustness of our findings by conducting sensitivity analyses that excluded patients with a Charlson Comorbidity Index score of 1 or higher to support the assumption that the upper gastrointestinal bleeding was not due to other causes, such as neoplasia, cirrhosis, or varicose veins.

Note: We have added one additional citation.

Text Changes (Materials and methods, Lines 322-325):

According to a multicenter, cross-sectional study conducted in 2022 in Japan, the mean age of new patients with hip osteoarthritis was 63.5 years [53], and those patients with osteoarthritis may undergo joint replacement surgery after taking NSAIDs for an extended duration.

 Additional Reference:

[53] Sato T, Yamate S, Utsunomiya T, Inaba U, Ike H, Kinoshita K, et al. Life Course Epidemiology of Hip Osteoarthritis in Japan: A Multicenter, Cross-Sectional Study. J Bone Joint Surg Am. 2024; Forthcoming. doi: 10.2106/JBJS.23.01044, PMID 38626018.

We thank Reviewer #1 and Reviewer #2 for the thorough review of our work, and provide comments to improve the quality of our paper.

---

## [Decision Letter · Decision Letter 2]

27 May 2024

PONE-D-23-40113R2Continuous co-prescription of rebamipide prevents upper gastrointestinal bleeding in NSAID use for orthopaedic conditions: A nested case-control study using the LIFE Study databasePLOS ONE

Dear Dr. Hamai,

Thank you for submitting your manuscript to PLOS ONE. After careful consideration, we feel that it has merit but does not fully meet PLOS ONE’s publication criteria as it currently stands. Therefore, we invite you to submit a revised version of the manuscript that addresses the points raised during the review process.

We look forward to receiving your revised manuscript.

Kind regards,

Mahmoud Kandeel

Academic Editor

PLOS ONE

Journal Requirements:

Reviewers' comments:

Reviewer's Responses to Questions

**Comments to the Author**

1. If the authors have adequately addressed your comments raised in a previous round of review and you feel that this manuscript is now acceptable for publication, you may indicate that here to bypass the “Comments to the Author” section, enter your conflict of interest statement in the “Confidential to Editor” section, and submit your "Accept" recommendation.

Reviewer #2: (No Response)

2. Is the manuscript technically sound, and do the data support the conclusions?

Reviewer #2: Yes

3. Has the statistical analysis been performed appropriately and rigorously? 

Reviewer #2: Yes

4. Have the authors made all data underlying the findings in their manuscript fully available?

Reviewer #2: Yes

5. Is the manuscript presented in an intelligible fashion and written in standard English?

Reviewer #2: Yes

6. Review Comments to the Author

Reviewer #2: Line 240.

I think what is meant is “The median time to upper gastrointestinal bleeding after the first prescription of NSAIDs was 134 days (S2 Figure).”

7. PLOS authors have the option to publish the peer review history of their article (what does this mean?). If published, this will include your full peer review and any attached files.

Reviewer #2: **Yes: **Javier Garjon

---

## [Author Response · Author response to Decision Letter 2]

27 May 2024

Response to Journal requirements:

Journal Requirements: Please review your reference list to ensure that it is complete and correct. If you have cited papers that have been retracted, please include the rationale for doing so in the manuscript text, or remove these references and replace them with relevant current references. Any changes to the reference list should be mentioned in the rebuttal letter that accompanies your revised manuscript. If you need to cite a retracted article, indicate the article’s retracted status in the References list and also include a citation and full reference for the retraction notice.

Response:

Thank you for the information. We re-searched all the articles on Pubmed and confirmed that they are complete and correct, and none have been retracted.

Response to Reviewer #2:

Reviewer #2: Line 240.

I think what is meant is “The median time to upper gastrointestinal bleeding after the first prescription of NSAIDs was 134 days (S2 Figure).”

Response:

Thank you so much for pointing out the error. Your revised description is correct. We also corrected the numbering of the figure; S1 is correct, not S2.

Text Changes (Results, Lines240-242):

The median time to upper gastrointestinal bleeding after the first prescription of NSAIDs was 134 days (S1 Figure).

We thank Reviewer #1 and Reviewer #2 for the thorough review of our work, and provide comments to improve the quality of our paper.

---

## [Editor Report · Decision Letter 3]

30 May 2024

Continuous co-prescription of rebamipide prevents upper gastrointestinal bleeding in NSAID use for orthopaedic conditions: A nested case-control study using the LIFE Study database

PONE-D-23-40113R3

Dear Dr. Hamai,

We’re pleased to inform you that your manuscript has been judged scientifically suitable for publication and will be formally accepted for publication once it meets all outstanding technical requirements.

Kind regards,

Mahmoud Kandeel

Academic Editor

PLOS ONE
---

## [Editor Report · Acceptance letter]

31 May 2024

PONE-D-23-40113R3 

PLOS ONE

Dear Dr. Hamai, 

I'm pleased to inform you that your manuscript has been deemed suitable for publication in PLOS ONE. Congratulations! Your manuscript is now being handed over to our production team.

Kind regards, 

on behalf of

Professor Mahmoud Kandeel 

Academic Editor

PLOS ONE